# Fabrication, Microstructure and Corrosion Resistance of Zn/Al Composite Coating by Arc Spraying

Bo Li [1], Dan Yang [2], Zhuoyi Liu [3], Jinhang He [1], Jie Bai [1], Haibo Jiang [4], Ye Tian [2], Zhiqing Zhang [2] and Shifeng Liu [2,*]

1   Electric Power Research Institute, Guizhou Power Grid Co., Ltd., Guiyang 550000, China; gzgylb2207@163.com (B.L.); jinhanghe1982@126.com (J.H.); bai_jie25@163.com (J.B.)
2   College of Materials Science and Engineering, Chongqing University, Chongqing 400044, China; yangdan9851@163.com (D.Y.); yyy0eee@163.com (Y.T.); zqzhang@cqu.edu.cn (Z.Z.)
3   Guizhou Electric Power Grid Dispatching and Control Center, Guiyang 550002, China; gylzy@sina.com
4   Duyun Power Supply Bureau, Guizhou Power Grid Co., Ltd., Duyun 561000, China; 18286498359@163.com
*   Correspondence: liusf06@cqu.edu.cn

**Abstract:** In this work, the Zn/Al composite coating was prepared on the surface of Q345 steel using arc spraying. The simple and efficient cold-pressing technique was used for the composite coating. The cold pressure sealing (CPS) technique is proposed to reduce the porosity of the composite coating. The corrosion behavior of Zn/Al composite coatings without and with cold pressure sealing treatment in a corrosive environment was studied. The microstructures of composite coatings without and with CPS were studied by scanning electron microscope (SEM), and the corrosion properties of composite coating without and with CPS were comparatively investigated. The results showed that the porosity of the composite coating was significantly reduced to 2% by CPS. After 28 days of immersion, the dense corrosion products filled the defect area in the CPS sample. After CPS treatment, the corrosion potential is $-0.829$ V, and the corrosion current density is $5.636 \times 10^{-6}$ A/cm$^2$. After cold pressing, the bonding strength of the coating is 13.82 MPa, which is 43% higher than that before the sealing treatment. The Zn/Al composite coating treated by CPS exhibits excellent corrosion resistance in the simulated marine environment.

**Keywords:** Zn/Al composite coating; cold pressure sealing treatment; corrosion resistance; porosity

## 1. Introduction

In order to protect the surface of steel substrate susceptible to environmental corrosion, the most commonly used method is to deposit a thin layer of corrosion-resistant material on the substrate, which is called coating [1]. Thermal spraying is one of the commonly used methods [2], which is a surface-strengthening technology for surface modification and protection of metal and alloy materials [3–5]. The technique uses a specific heat source to heat the spray material to a molten state, and the molten droplets are accelerated by external airflow and then sprayed onto the pretreated substrate surface, and finally, the coating of a certain thickness is formed on the substrate surface. In recent years, arc spraying technology has been widely used to prepare protective coatings on low carbon steel or low alloy steel due to its simple operation, low cost, ability to spray different materials and its use in large steel structures, such as ships, bridges and offshore platforms [6–8].

Generally speaking, the contact process between the coating and the substrate after thermal spraying atomization is collision → coating particle flattening → metal condensation → stratification [9]. When the metal forms the coating on the surface of the substrate, the coating has a layered structure. In the spraying process, the particles in the molten state impact the deposited particles and produce the "shielding effect"; the molten particles become solid after cooling, the volume shrinkage without excess molten particles supplement and the gas between particles cannot be completely discharged; incomplete

overlap between particles and other factors lead to the inevitable pores [10]. In a certain corrosive environment, due to the presence of porosity, the corrosive medium is in contact with the matrix and accelerates the corrosion rate, thus reducing the corrosion resistance of the coating [11]. Therefore, in practical application, it is necessary to conduct subsequent sealing treatment on the sprayed coating to reduce the porosity and to improve the corrosion resistance of the coating [12,13]. The methods commonly used for coating sealing treatment include sealing agent treatment, laser remelting, heat treatment, etc. Pang et al. [14] used cathodic electrophoresis to seal the arc-sprayed Al coating with epoxy resin. The full immersion experiment was conducted in 3.5 wt% NaCl solution. The results showed that the pores are completely covered by the epoxy resin with a thickness of about 20 um, and the surface roughness of Al coating is reduced. After corrosion, the thickness of the epoxy resin layer does not change and can still be completely covered on the sprayed Al coating. Zhang et al. [15] deposited $Cr_2O_3$-8%$TiO_2$ coating on low carbon steel by plasma spraying, sealed with epoxy resin and silicone resin in different environments, and studied the microstructure and corrosion resistance of the coating without and with sealing. The results show that the silicone sealing agent can seal the coating well, most of the pores are blocked, and the sealant can penetrate into the pore. The coatings sealed with silicone have the highest corrosion potential and the lowest corrosion current density under vacuum conditions. KIM et al. [16] applied fluorosilicone sealant at room temperature several times to improve the marine corrosion resistance of Al-Zn coating, and the coated sealing sample had a lower corrosion current density. Zhang et al. [17] prepared Al-Fe-Nb-Ni coating on low-carbon steel by arc spraying and then sealed the coating surface by laser remelting. It was found that after remelting, the porosity of the coating was reduced to 2%, the compactness was improved, and metallurgical bonding was formed at the interface between the substrate and the coating. Ji et al. [18] chose a laser treatment instead of a sealing treatment and compared the cross-section morphology of coating without and with laser treatment. The sprayed coating is a typical layered structure with certain pores. After laser treatment, the coating microstructure becomes more compact, the lamellar structure disappears, and only a few pore defects exist. In summary, the above methods can reduce porosity, but there are some shortcomings. For example, the time required for sealing agents and heat treatment is longer (sealant treatment), and laser remelting sealing has higher requirements for equipment and working environment. If it is a small laser spot, it needs to use multiple-lap technology to ensure accuracy. In addition to the sealing treatment of Zn/Al coating with a sealing agent, the Al/Zn alloy coating has a good plastic deformation ability, in view of the pores formed in the arc spraying process, the pores and defects of the coating can be improved by plastic deformation [16]. By cold pressing the coating, the surface structure of the coating is changed to reduce the porosity of the coating and achieve a good sealing treatment. Therefore, the cold-pressing sealing process of Zn/Al coating is expected to be a simple and effective sealing method. However, there are few studies on reducing the porosity of the coating by direct cold pressing [19].

In the present work, the Zn/Al composite coating was prepared on the surface of Q345 carbon steel by arc spraying. A simple and efficient cold pressure sealing technique was used to seal the treatment. The microstructure and corrosion resistance of the coating without and with cold pressure sealing were studied.

## 2. Materials and Methods

### 2.1. Materials and Coating Preparation

The substrate was common steel (Q345, Chongqing Iron and Steel Co., Ltd, Chongqing, China) in this work, and the chemical composition is shown in Table 1. It is machined to a size of 10 mm × 10 mm × 5 mm using wire cutting. The material wire used for arc spraying is pure Al wire and pure Zn wire with a diameter of 1.6 mm. The sample was ultrasonically cleaned before arc spraying, then derusted by laser cleaning and sandblasted. All coatings are prepared on Q345 substrate using an automatic laser derusting and spraying system. The specific spraying parameters are as follows: spraying voltage (29 V), spraying current

(180 A), spraying distance (180 mm) and scanning rate (0.01 m/s). The preparation principle of the Zn/Al composite coating is as follows: firstly, the certain thickness of Zn coating is prepared on the surface of the substrate, and after the Zn coating is sprayed and the Al coating is prepared on the surface of the Zn coating. As a result, a Zn/Al composite coating with Zn coating on the bottom and Al coating on the surface is formed as shown in Figure 1.

**Table 1.** Q345 composition information (mass fraction, wt%).

| Element | Fe | Mn | Si | Cr | Cu | Ni | Al | S | P |
|---|---|---|---|---|---|---|---|---|---|
| (wt%) | 97.76 | 1.59 | 0.35 | 0.10 | 0.09 | 0.04 | 0.03 | 0.02 | 0.02 |

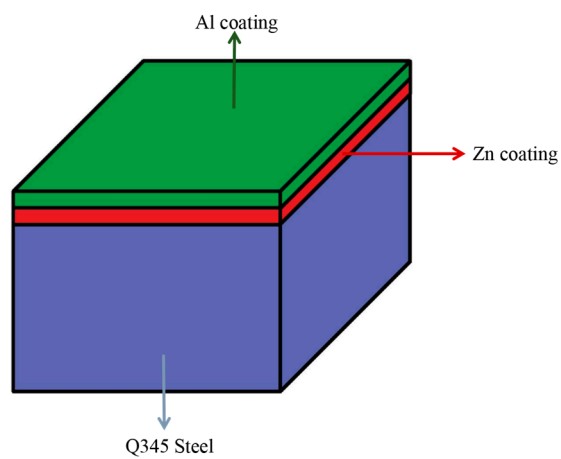

**Figure 1.** Schematic diagram of Zn/Al composite coating.

### 2.2. Cold Pressure Sealing Treatment of Composite Coating

For the Zn/Al composite coating, a Xinsansi CMT-5105 (Chongqing, China) universal electronic experimental machine was used to carry out a cold-pressing sealing treatment for composite coating. The cold-pressing parameters were 450 MPa for 60 s.

### 2.3. Roughness and Porosity Analysis

The OLS4000 laser confocal scanning microscope (Biotimes Technology Limited, Shanghai, Chian) was used to observe and analyze the roughness state of the coating surface without and with CPS, which is helpful to analyze the influence of roughness change on the corrosion resistance of the coating. Three different locations of each sample were selected to calculate the roughness value, and finally, the average value was obtained. Image J software (v1.8.0, National Institutes of Health, Stapleton, NY, USA) was used to analyze the porosity of composite coatings without and with CPS. According to the pore size in the picture, the porosity value is obtained.

### 2.4. Corrosion and Electrochemical Experiments

The NaCl solution was used to simulate the marine environment for full immersion. In this paper, 3.5 wt% NaCl solution was selected, the size of the sample was 15 mm × 15 mm × 5 mm, and the uncoated layer was covered with rubber. The sample was immersed in the solution completely, and the experimental temperature was kept at about 35 °C by using a water bath. The immersed time was 7 days, 14 days, and 28 days. Then, the morphology characterization, composition analysis and potentiodynamic polarization test of the immersion samples with different times were carried out. The potentiodynamic polarization curve and electrochemical impedance spectrum of the composite coating were measured using a Princeton electrochemical workstation(Princeton, NJ, USA). The sample size was 15 mm × 15 mm × 5 mm. The common three-electrode system was used for electrochemical measurement, with reference electrode saturated

calomel electrode and auxiliary electrode Pt electrode, and the sample was used as the working electrode. The electrolyte solution was 3.5 wt% NaCl solution, and the exposed area of the sample was 1 cm$^2$. In a polarization curve test, the potential scanning range was based on the open-circuit potential $\pm$500 mV, and the scanning speed was 1 mV/s. The electrochemical impedance spectrum was tested at an open-circuit potential. During the test, the AC voltage disturbance value was set at 10 mV, and the test frequency range was $10^{-2}$ to $10^5$ Hz. After the test, Corrview software (Corrview, Landing, NJ, USA) was used to fit and analyze the polarization curve. The morphology and composition of the composite coating without and with cold pressing were analyzed by scanning electron microscopy (SEM, JEOL 7800 FEG, Tokio, Japan) and energy dispersive spectroscopy (EDS).

### *2.5. Bonding Strength*

The coating bonding strength was tested in accordance with GB /T 8642 [20], and the testing equipment was an electronic universal testing machine as shown in Figure 2. The test consists of a two-part loading block and a square sample (10 mm $\times$ 10 mm). Then the sample and the two-part loading blocks were glued together with A/B adhesive. The parts of the sample assembly remain vertical in the gripper and the sample is placed for a period of time until the binder is fully cured. The strength RH (N/mm$^2$) obtained by the test is calculated by the quotient of the maximum load F (N) and the cross-sectional area S (mm$^2$) of the fracture surface.

$$R_H = F/S \tag{1}$$

The multiple tests were carried out for the strength test.

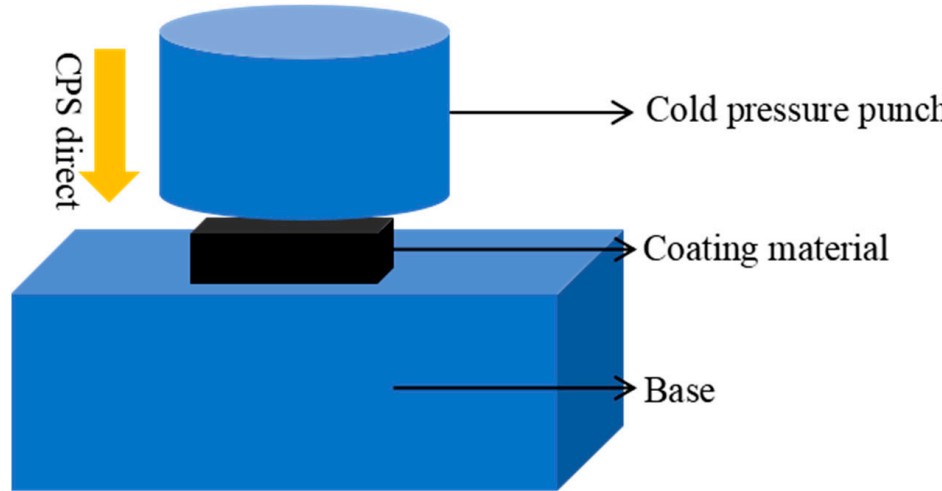

**Figure 2.** Schematic diagram of CPS.

### 3. Results and Discussions

#### *3.1. Macroscopic Morphology without and with CPS*

Figure 3 shows the macroscopic morphology and roughness of Zn/Al composite coating without and with CPS. Figure 3a shows the sprayed surface of Zn/Al coating after arc spraying. There are no obvious macroscopic cracks and defects on the surface, but the surface roughness is relatively high. Figure 3b shows the surface after the cold pressure sealing treatment. Under the action of external pressure, semi-molten or un-molten particles are extruded into the flat shape, forming an obvious extrusion trail. The pores in the composite coating can be effectively supplemented after CPS, greatly reducing the roughness of the coating surface and forming a smoother, more compact and more uniform surface [21].

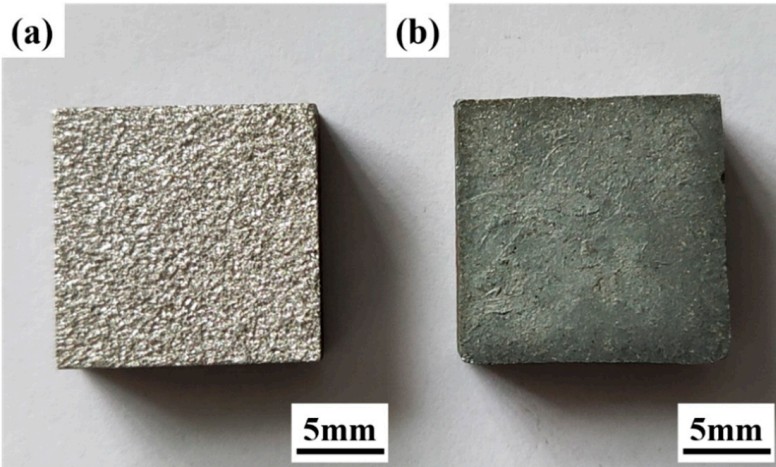

**Figure 3.** Schematic diagram of Zn/Al composite coating: (**a**) without CPS; (**b**) with CPS.

### 3.2. Microstructure and Energy Spectrum Analysis

Figure 4 shows the microstructure of the Zn/Al composite coating. As can be seen from the figure, the upper part is divided into an Al coating, the middle is a Zn coating, and the bottom is Q345. Though the Zn coating and Q345 have similar colors, they can still be clearly distinguished at the interface. The energy spectrum analysis of the longitudinal section of the substrate coating shows that the Zn coating at the bottom has good contact with the substrate, and no cracks are found. The interface between the Zn coating and Al coating is a good contract. At the interface of the Zn/Al composite coating, the molten Al is embedded on the surface of the Zn coating so that the surface pores of the Zn coating are filled, forming a bonding interface as shown in the white dashed line. At the same time, an oxide film of a certain thickness (about tens of microns) is formed on the surface of the Al coating. No holes, cracks or other defects were observed in the entire section. It can be seen from the figure that a good quality Zn/Al composite coating can be obtained by arc spraying. Based on this, the follow-up experiment is carried out.

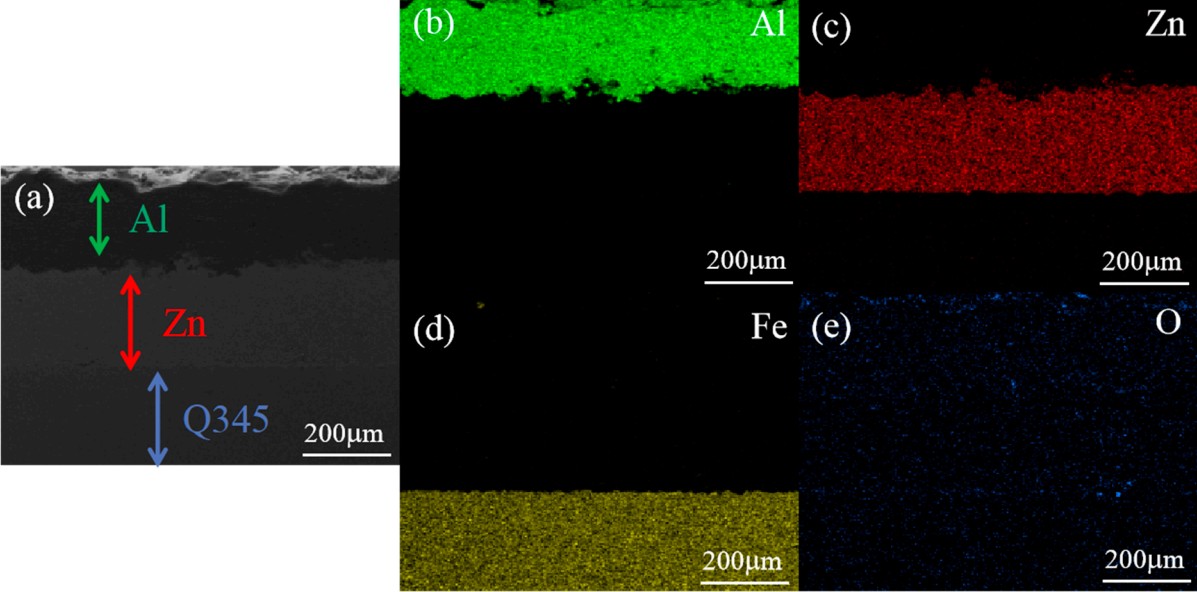

**Figure 4.** Morphology and EDS images of composite coating without cold pressing (**a**) longitudinal section morphology, (**b**) Al element distribution, (**c**) Zn element distribution, (**d**) Fe element distribution, (**e**) O element distribution.

### 3.3. Roughness and Porosity Analysis

It can be seen from Figure 3b that after CPS, the surface roughness of the composite coating is significantly reduced. The rougher the surface of the coating, the easier the corrosive medium is to gather, and the worse the anti-corrosion effect. In order to further analyze the change of roughness, an OLS4000 laser confocal scanning microscope was used to observe the roughness of the coating. Figure 5 is the comparison image of roughness without and with CPS. It can be seen from the figure that the roughness value (Ra) of the coating surface without and with CPS is about 22 μm and about 2.1 μm, respectively. After CPS, the surface roughness of the coating is significantly reduced; it can be obviously compared by different scales. The reduced roughness reduces the contact time of the corrosive medium on the coating surface and is conducive to reducing pitting, thereby improving corrosion resistance.

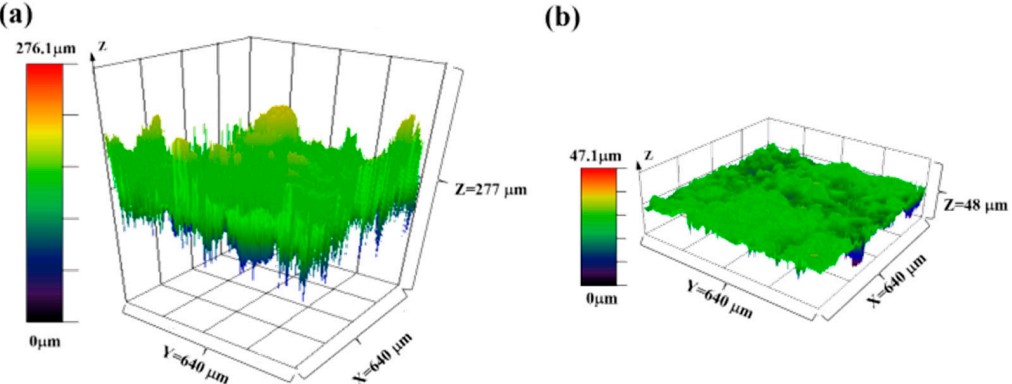

**Figure 5.** Schematic diagram of roughness: (**a**) roughness without CPS; (**b**) roughness with CPS.

By comparing Figure 3, it can be seen that the surface porosity of the coating was significantly reduced after the sealing. For the samples after cold pressing, due to the relatively difficult internal pore statistics, only the surface pore statistics were carried out. The Image J was used to make statistics on the porosity of the coating without and with CPS, and the results were shown in Figure 6. Image J software, the void is marked in red, and the porosity can be estimated by the ratio of the red area to the whole. The figure shows that the surface porosity of the coating without and with CPS is about 9.8% and about 2.4%, respectively. This indicates that the porosity of the composite coating can be significantly reduced by CPS, and the coating with better surface quality can be obtained, which reduces the surface area and improves the surface density of the coating, which is conducive to preventing the infiltration of corrosive media and improving the anti-corrosion performance.

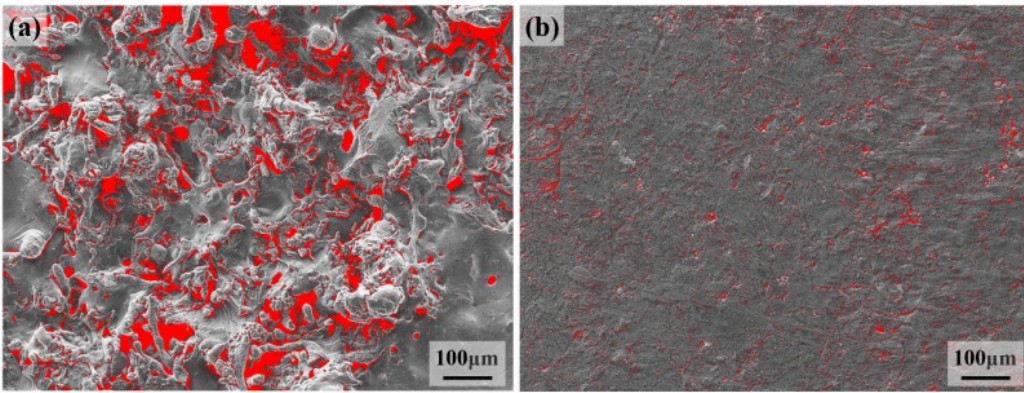

**Figure 6.** Porosity of composite coating: (**a**) without CPS; (**b**) with CPS.

### 3.4. Morphology Analysis of Corrosion Products

3.4.1. Macroscopic Morphology Analysis

Figure 7 shows the macroscopic corrosion morphology of the Zn/Al composite coating without and with CPS after being immersed in a 3.5% NaCl solution for 7 days, 14 days and 28 days. For the sample without CPS, with the extension of the immersion time, white spot-like corrosion products almost covered the entire surface of the coating. After CPS, there are a small number of corrosion products on the edge of the coating surface after immersion for 7 days. The corrosive medium is easy to accumulate on the edge of the coating, promoting the generation of corrosion products. After 14 days of immersion, a small number of corrosion products were produced on the surface, and the corrosion products were dense. After 28 days of immersion, the amount of corrosion products on the surface was still small, and only some areas of corrosion products appeared. Generally speaking, compared with the coating without the sealing treatment, the corrosion products did not completely cover the entire surface of the coating during the immersion process, and the positions where the corrosion products were produced were generated by the extrusion of various particles after cold pressing. Therefore, the composite coating with CPS is corroded lightly; the corrosion products did not enter the matrix so the corrosion resistance of the cold pressing hole-sealing coating is improved.

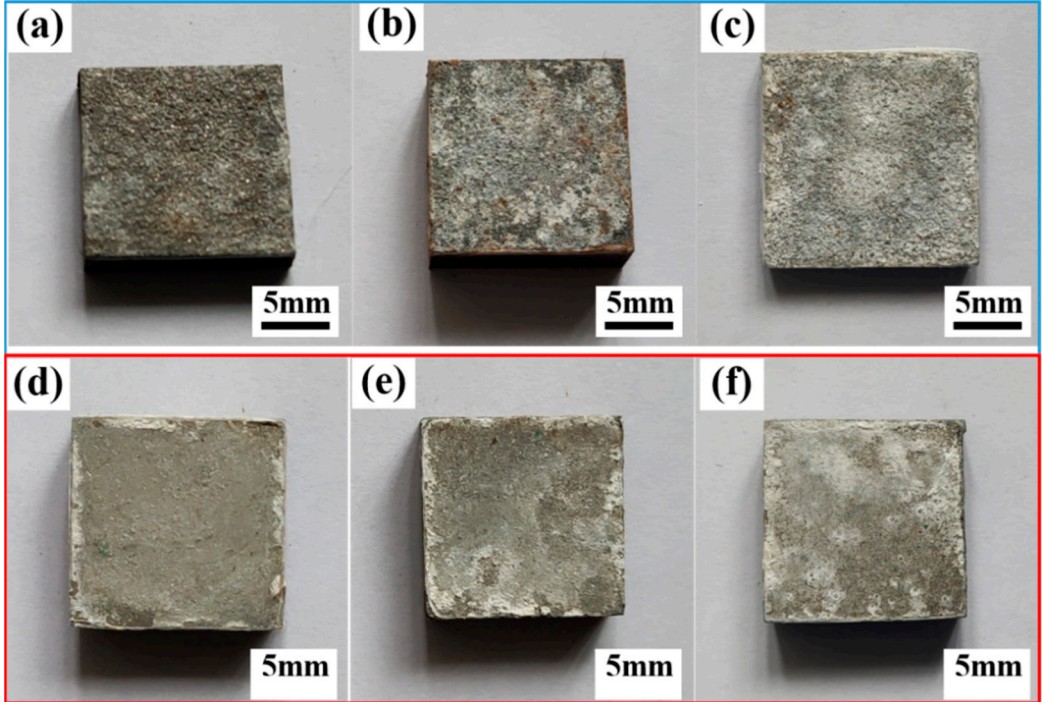

**Figure 7.** Macro-corrosion morphology of composite coating under different immersion time (**a**) 7 days without CPS, (**b**) 14 days without CPS, (**c**) 28 days without CPS, (**d**) 7 days with CPS, (**e**) 14 days with CPS, (**f**) 28 days with CPS.

3.4.2. Microstructure Analysis

Figure 8 shows the images of composite coating without and with CPS under different immersion times. It can be seen from the figure that after the cold pressing hole-sealing treatment, there is some corrosion on the surface of the composite coating at the early immersion stage, and only white pitting particles appear, which are discontinuous as shown in Figure 8d. After 14 days of immersion, a large number of cracks and other defects appeared on the surface of the coating as shown in the red circles of Figure 8e, and the corrosion products precipitated and accumulated along the defects. After immersion for 28 days, the corrosion morphology of the coating is similar to that of immersion for 14 days;

no more serious corrosion occurs, but the corrosion products are more compact, from loose flocculent to more compact compared with Figure 8c. At the same time, after the cold pressing sealing hole, the corrosion products precipitated along the cold pressing trace line gap as shown by the red dashed line in Figure 8d. After the cold pressing corrosion, the dense products formed also partially filled the defective part, further strengthening the corrosion resistance. The morphology of the un-immersion sample is similar to the macroscopic structure, so it is not discussed further.

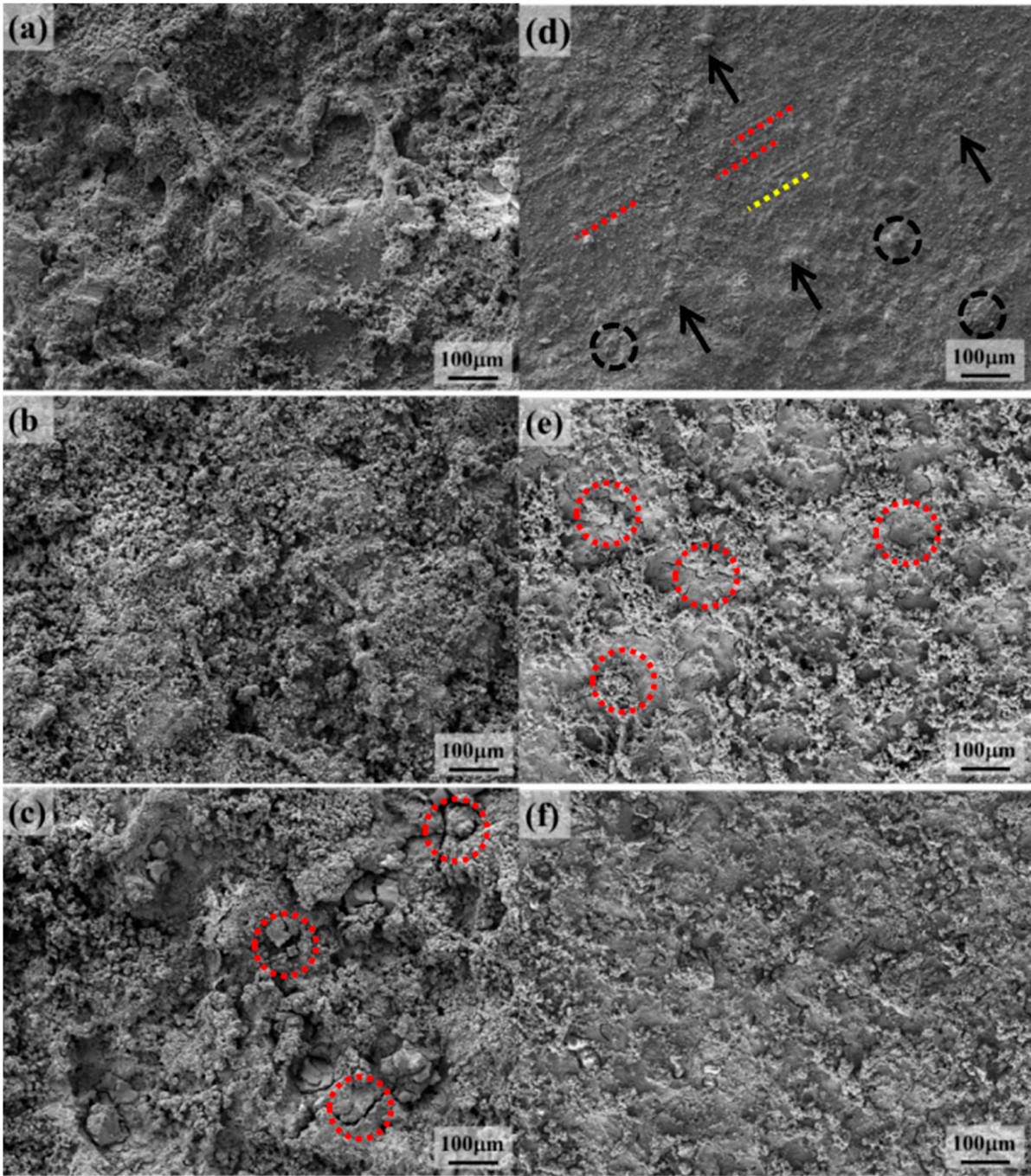

**Figure 8.** Micro-corrosion morphology of composite coating under different immersion time (**a**) 7 days-without CPS, (**b**) 14 days-without CPS, (**c**) 28 days-without CPS, (**d**) 7 days-with CPS, (**e**) 14 days-with CPS, (**f**) 28 days-with CPS.

### 3.5. Bonding Strength Test of Cold Pressing Sealing Coating

The bonding strength of the coating includes the bonding strength between the coating and the substrate and the bonding strength between the particles inside the coating, which is an important index of the mechanical properties of the coating. Good bonding strength is the important factor that the coating provides long-term protection for the substrate. In general, the failure of the coating generally occurs between the coating and the substrate, because the physical properties of the coating and the substrate are different, and there is a temperature difference. During arc spraying, when the molten droplets hit the surface of the cold substrate, they will be chilled and solidified, resulting in microscopic shrinkage stress, and the accumulation of stress will cause residual stress of the coating as a whole, thus affecting the service life of the coating. In order to improve the bonding strength of the coating, the morphology of the substrate can be changed by sandblasting. After sandblasting, the roughness and surface area of the substrate surface increase, which improves the bonding strength of the coating to a certain extent. However, the pores between the coating and the substrate are still unavoidable by arc spraying [11]. The cold pressing process not only compacts the coating, making the interior of the coating more dense as show in Figure 5, but also helps the coating to be embedded in the substrate to form mechanical pinning, thus further increasing the bonding strength between the coating and the substrate. As can be seen from Table 2, the strength of composite coating after the cold pressure treatment is 13.82 MPa, which is 43% higher than that before cold pressing. Each set of tests was performed three times and then averaged.

**Table 2.** Bonding strength after CPS of Zn/Al composite coating.

| Materials | Bonding Strength (MPa) |
|---|---|
| Without CPS | 9.66 |
| With CPS | 13.82 |

### 3.6. Electrochemical Performance Test

Figure 9 shows the polarization curves of the Zn/Al composite coating without and with the cold pressure sealing treatment. The polarization curve is divided into two parts: the upper part corresponds to the destruction and dissolution reaction of the coating, and the lower part corresponds to the process of hydrogen evolution by water reduction [15]. As can be seen from the figure, the polarization curve of the coating moved to the upper after the cold pressure sealing treatment, indicating that the corrosion potential was positive and the corrosion current density was lower. Table 3 shows the corresponding electrochemical parameters obtained by Tafel fitting. As can be seen from the table, the corrosion potential of the coating is $-0.829$ V after the cold pressure sealing treatment, greater than the potential before the cold pressure sealing treatment ($-1.084$ V), and the corrosion potential can evaluate the corrosion tendency of the material. Generally speaking, the more positive the corrosion potential value is, the more difficult the corrosion of the material is. At the same time, the corrosion current density of the coating is $5.636 \times 10^{-6}$ A/cm$^2$ after CPS, which is about an order of magnitude lower than before sealing treatment, so the corrosion rate is lower, indicating that cold pressure sealing treatment is conducive to improving the corrosion resistance of the coating. Compared with the unsealed composite zinc-aluminum coating, the corrosion resistance has been greatly improved [22–24]. The excellent corrosion resistance can be attributed to the fact that the surface defects such as micro-pores and cracks can be effectively filled after the cold pressure sealing treatment of the rough surface of the spray coating, and the surface roughness is greatly reduced so that the composite coating can form a more stable and dense oxide film on the surface as shown in Figure 6, so as to show a corrected corrosion potential. In addition, due to the reduction of initial pores, the corrosive medium has fewer channels to enter the coating, and the corrosion resistance increases, so the corrosion current density is lower after cold pressing.

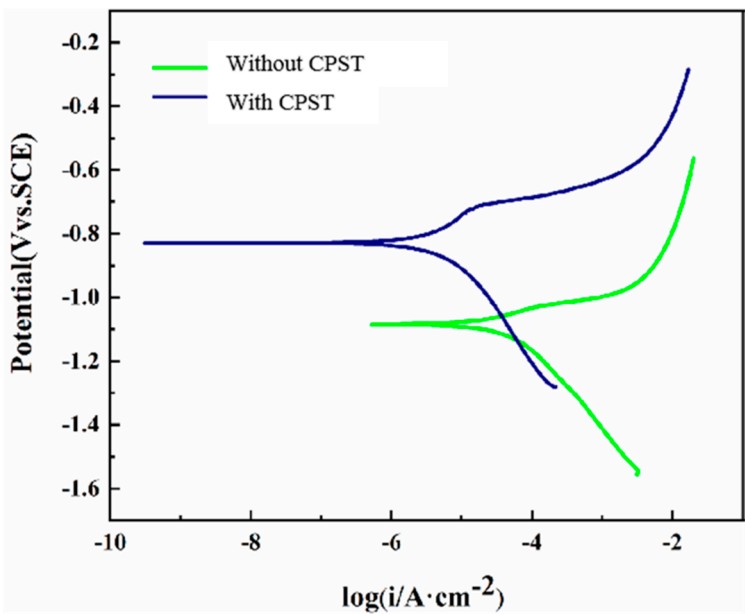

**Figure 9.** Polarization curves of composite coating without and with cold pressure sealing treatment.

**Table 3.** Electrochemical parameters of composite coating without and with CPS.

| Sample | Corrosion Voltage (V) | Corrosion Current Density (A/cm$^2$) |
| --- | --- | --- |
| Without CPS | −1.162 | $3.445 \times 10^{-5}$ |
| With CPS | −0.829 | $5.636 \times 10^{-6}$ |

In order to further understand the corrosion behavior of the composite coating after the cold pressure hole sealing treatment, the electrochemical impedance spectroscopy test was carried out to further the detailed information of the corrosion process so as to obtain the corrosion mechanism of the coating. Figure 10 shows Nyquist and Bode diagrams of composite coating without and with CPS. The composite coating still presents three time constants after CPS. The capacitance at high frequency is related to the double electric layer at the interface between the Al coating and corrosion solution, the capacitance at the middle frequency is related to the internal degradation of the coating, and the capacitance at the low frequency is related to the substrate dissolution reaction [19]. The diameter of the capacitive reactance arc can directly reflect the corrosion resistance of the material. The larger the diameter of the capacitive reactance arc, the better the corrosion resistance. It can be seen from the Nyquist diagram that the capacitance frequency of the coating after CPS is larger, indicating that the corrosion resistance of the coating after CPS is greatly improved. Figure 10b,c demonstrates this situation from the side without further analysis.

In order to further analyze the electrochemical characteristics of the coating and its structural characteristics, the equivalent circuit model as shown in Figure 10 was obtained by fitting the impedance data. Figure 11 is a schematic diagram of the simulation. The fitting parameters of each equivalent element are shown in Table 4. ($\chi^2$) of the fitting error value can be used to assess whether the fitting condition is good. Generally speaking, if the error value is less than $10^{-3}$, it is considered that the fitting degree between the test data and the equivalent circuit is good. Different components in the equivalent circuit represent different components of the actual coating. A constant phase component CPE (denoted as Q) is used to replace the capacitor, where Rs is the corrosion solution resistance, R$_{out}$ and Q$_{out}$ are the resistance and capacitance of the surface layer, which can be used to detect the shielding effect of the coating on the corrosive medium and evaluate the anticorrosive performance of the surface coating [25]. R$_{in}$ and Q$_{in}$ are the interface resistance and capacitance of the Zn/Al coating, R$_{ct}$ is the charge transfer resistance when the substrate dissolves, and Q$_{dl}$

is the double-layer capacitance at the substrate interface. No further analysis was made without CSP.

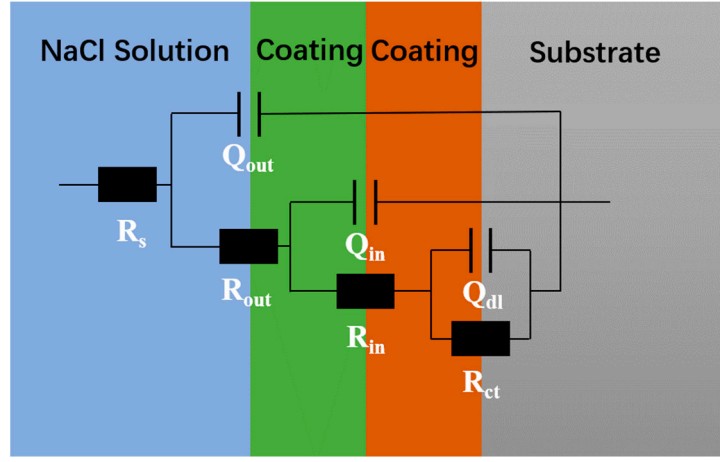

**Figure 10.** EIS of composite coating without and with CPS: (**a**) Nyquist, (**b**) |Z| mode value, (**c**) Phase diagram.

**Figure 11.** EIS equivalent circuit diagram after CPS of composite coating.

**Table 4.** EIS parameters with CPS of composite coating.

| Equivalent Element / Sample | $R_{out}$ ($\Omega \cdot cm^2$) | $Q_{out}$-$Y_0$ ($S \cdot s^n \cdot cm^{-2}$) | n1 | $R_{in}$ ($\Omega \cdot cm^2$) | $Q_{out}$-$Y_0$ ($S \cdot s^n \cdot cm^{-2}$) | n2 | $R_{ct}$ ($\Omega \cdot cm^2$) | $Q_{dl}$-$Y_0$ ($S \cdot s^n \cdot cm^{-2}$) | n3 | $\chi^2$ |
|---|---|---|---|---|---|---|---|---|---|---|
| After CPS | 5257 | $3.24 \times 10^{-5}$ | 0.80 | 2512 | $1.04 \times 10^{-4}$ | 0.71 | 2834 | $6.34 \times 10^{-4}$ | 0.68 | $3.47 \times 10^{-4}$ |

Generally speaking, the corrosion resistance of the coating can be evaluated by the total resistance value of each resistance element after fitting. The total resistance value is usually negatively correlated with the corrosion rate. The higher the total resistance value of the coating, the slower the corrosion process will be. As can be seen from Table 4, the total resistance of the composite coating after CPS is 10,603 $\Omega \cdot cm^2$, which is greater than the total resistance of the coating before CPS (3667.2 $\Omega \cdot cm^2$), and its corrosion resistance is greatly improved. This is because in the corrosive medium where $Cl^-$ exists, $Cl^-$ destroys the oxide film very quickly [19]. Before CPS, there are many pores on the surface of the composite coating, which make the corrosive medium easily accumulate inside the coating, and eventually lead to the formation of holes, which the corrosive medium penetrates into the interface between the coating and the substrate. After sealing the hole, the semi-molten or unmolten particles on the surface of the composite coating are spread out due to the external pressure, and the original pores are filled and become strip-shaped gaps, and these gaps become shallow because of extrusion. In the corrosion process, corrosion products are easy to accumulate in the gap and prevent the corrosive medium from continuing to penetrate. Therefore, the cold-pressed hole can improve the corrosion resistance of the composite coating and greatly extend the service life of the coating.

## 4. Conclusions

The single Zn coating or Al coating has obvious structural shortcomings when it is used as an anti-corrosive surface coating, such as high surface roughness and pores. In order to better play the anti-corrosive advantages of composite coating, the composite coating is treated with a cold pressure sealing to change the structural characteristics. The microstructure morphology and corrosion resistance of the composite coating without and with CPS were studied. The main research results are as follows:

(1) After the cold pressure sealing treatment, the roughness of the coating surface is reduced. The roughness of the composite coating is reduced from about 22 μm to about 2.1 μm, which is about 90%. Under the action of pressure, the internal porosity of the coating is reduced, and the porosity of the coating is reduced from about 9.8% to about 2.4%.

(2) After CPS, the coating and substrate were embedded to form mechanical nailing, thus increasing the bonding strength of the coating and substrate, and the bonding strength of the composite coating reached 13.82 MPa.

(3) The more stable and dense surface was formed after CPS on the surface of the composite coating with the extension of immersion time, and the coating with CPS showed the corrected corrosion potential ($-0.829$ V), lower corrosion current density ($5.626 \times 10^{-6}$ A) and better impedance value.

(4) The existing defects are filled with dense corrosion products, further blocking the corrosion channel, so the CPS can improve the corrosion resistance of the composite coating.

**Author Contributions:** Conceptualization, S.L.; Methodology, D.Y., Z.L., J.H., J.B. and H.J.; Validation, Z.Z.; Investigation, Y.T.; Resources, B.L. All authors have read and agreed to the published version of the manuscript.

**Funding:** This research was financially supported by the Science and technology project of China Southern Power Grid Co., Ltd. (Grant No. GZKJXM20191302).

**Institutional Review Board Statement:** Not applicable.

**Informed Consent Statement:** Not applicable.

**Data Availability Statement:** The data presented in this study are available on request from the corresponding author.

**Conflicts of Interest:** The authors declare no conflict of interest.

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
