# Peer review of "Fabrication, Microstructure and Corrosion Resistance of Zn/Al Composite Coating by Arc Spraying"

_coatings, doi:10.3390/coatings13081406_

Round 1

Reviewer 1 Report

This work deal with the possibility of creating a metal- based Zn/Al composite coating on steel substrates employing the arc spray technology. The aim of the work is interesting as it proposes a method to improve the corrosion protection characteristic of the coating created. However, some issues with this work should be addressed.

- In line 28 of the introduction the authors refer to "thermal spray technology" as a method to create coatings. However, this is not correct as thermal spray technologies are a "family" of different processes including Plasma spraying, Cold Spray, Wire-arc spraying and so on. Moreover, t not all the thermal spray techniques require the melting of the feedstock material.

-  It is not clear why the steel substrate is referred as the "Matrix". The substrate beneath the coating, in fact, is not a composite so the term is not correct. The matrix is the continuous constituent of a composite material in which the other phase is dispersed. In this case, it doesn't seem that the substrate suits the definition.

-Some further information about the cold pressure treatment, which is the focus of the paper, should be added

-Does the procedure followed to simulate marine environment refers to any standard test procedure?

- Reference [19] should refer to the test procedure and not another work.

Overall the work is methodologically acceptable and the proposed analyses are quite interesting. I suggest the acceptance after minor revisions.

The work is well written with the exception of minor errors and typos.

Author Response

Comments and Suggestions for Authors

This work deal with the possibility of creating a metal- based Zn/Al composite coating on steel substrates employing the arc spray technology. The aim of the work is interesting as it proposes a method to improve the corrosion protection characteristic of the coating created. However, some issues with this work should be addressed.

- In line 28 of the introduction the authors refer to "thermal spray technology" as a method to create coatings. However, this is not correct as thermal spray technologies are a "family" of different processes including Plasma spraying, Cold Spray, Wire-arc spraying and so on. Moreover, t not all the thermal spray techniques require the melting of the feedstock material.

This section has been modified to describe thermal spraying. Thermal spraying is one of the methods of coating preparation.

-  It is not clear why the steel substrate is referred as the "Matrix". The substrate beneath the coating, in fact, is not a composite so the term is not correct. The matrix is the continuous constituent of a composite material in which the other phase is dispersed. In this case, it doesn't seem that the substrate suits the definition.

Thank you very much for the revision of the matrix. After consideration, we think that the substrate is more consistent with the matrix material, and modify the matrix into the matrix material.

-Some further information about the cold pressure treatment, which is the focus of the paper, should be added.

For cold pressing treatment experiment, we are based on the evolution of isostatic pressing experiment, so there are few reports on cold pressing coating. The specific experimental process is as follows: For Zn/Al composite coating, Xinsansi CMT-5105 universal electronic experimental machine was used to carry out cold pressing sealing treatment for composite coating. The cold pressing parameters were: 450 MPa for 60s.

-Does the procedure followed to simulate marine environment refers to any standard test procedure?

The simulation of the marine environment was carried out in accordance with the standards, and a large number of references were made. Finally, 3.5 wt. % NaCl solution was selected.

- Reference [19] should refer to the test procedure and not another work.

The binding force was tested according to the standard test procedure, which is also referred to in reference 19. The test standard is GB/T 8642.

Overall the work is methodologically acceptable and the proposed analyses are quite interesting. I suggest the acceptance after minor revisions.

Reviewer 2 Report

Fabrication, microstructure and corrosion resistance of Zn/Al composite coating by arc spraying

The manuscript deals with the analysis of cold pressure sealing on Zinc – Aluminum depositions on steel substrates. The authors compare the coating as deposited and the coating treated by cold pressure sealing by visual inspection, roughness analysis, image analysis, response to a corrosive environment, and electrochemical analysis.

The topic and the analyses conducted are worth the interest of the readership of Coatings. The discussion should be enriched. The graphical aspect and the English language can be improved.

  • The discussion of the observed/measured morphology should be enriched. The data collected by confocal microscopy can be used to measure roughness and surface parameters, which could add high value to the manuscript.
  • Figure 1 should be remarkably improved. In Figure 4 it could be beneficial to use the same color scale for the two surfaces, in order to evidence the difference due to the cold pressure sealing.
  • Going through the text, I detected some minor typos or language miswriting. Please run a grammar check.
  • Going through the text, I detected some minor typos or language miswriting. Please run a grammar check.

Author Response

Comments and Suggestions for Authors

Fabrication, microstructure and corrosion resistance of Zn/Al composite coating by arc spraying

The manuscript deals with the analysis of cold pressure sealing on Zinc – Aluminum depositions on steel substrates. The authors compare the coating as deposited and the coating treated by cold pressure sealing by visual inspection, roughness analysis, image analysis, response to a corrosive environment, and electrochemical analysis.

The topic and the analyses conducted are worth the interest of the readership of Coatings. The discussion should be enriched. The graphical aspect and the English language can be improved.

  • The discussion of the observed/measured morphology should be enriched. The data collected by confocal microscopy can be used to measure roughness and surface parameters, which could add high value to the manuscript.
  • Figure 1 should be remarkably improved. In Figure 4 it could be beneficial to use the same color scale for the two surfaces, in order to evidence the difference due to the cold pressure sealing.

Thank you for your suggestion. Figure 1 has been replaced. Figure 5 clearly shows the difference before and after treatment. As can be seen from Fig. 5b, after cps, the average value under this scale is about 20. Compared with the average value of 130 before cps, it is significantly reduced. Although the scale is different, there is a significant reduction

  • Going through the text, I detected some minor typos or language miswriting. Please run a grammar check.

Thank you for your suggestion. A grammar check was carried out throughout the full text. Some changes have been made to the grammar and language of the whole paper.

Reviewer 3 Report

The article under review is devoted to the study of the microstructure and corrosion properties of Zn/Al composite coating obtained by arc spraying. The aim of the work is to evaluate the effect of cold pressure sealing on the microstructure and corrosion resistance.The paper describes the research methods in detail and clearly. Modern scientific equipment is used. Photo and graphic material of high quality.Scientific results are explained in detail and clearly.

Conclusions based on the research results have specific numerical values, practical significance and scientific novelty. In the abstract, in multiples, but with numerical values, the main results of the study are given.

The article is well written and a pleasure to read.The reference contains 24 articles, 11 of which are from the last 3 years.

Author Response

Comments and Suggestions for Authors

The article under review is devoted to the study of the microstructure and corrosion properties of Zn/Al composite coating obtained by arc spraying. The aim of the work is to evaluate the effect of cold pressure sealing on the microstructure and corrosion resistance.The paper describes the research methods in detail and clearly. Modern scientific equipment is used. Photo and graphic material of high quality.Scientific results are explained in detail and clearly.

Conclusions based on the research results have specific numerical values, practical significance and scientific novelty. In the abstract, in multiples, but with numerical values, the main results of the study are given.

The article is well written and a pleasure to read.The reference contains 24 articles, 11 of which are from the last 3 years.

Thanks very much!

Reviewer 4 Report

Authors must improve the manuscript as suggested by the reviewer in the attached pdf file. Thereafter, it may be reconsidered for publication. 

Please see the relevant comments at the appropriate places in the pdf file. In general, complex sentences have to be reconstructed to clarify the scientific ideas presented in the manuscript.

Author Response

Comments and Suggestions for Authors

Authors must improve the manuscript as suggested by the reviewer in the attached pdf file. Thereafter, it may be reconsidered for publication.

Comments on the Quality of English Language Please see the relevant comments at the appropriate places in the pdf file. In general, complex sentences have to be reconstructed to clarify the scientific ideas presented in the manuscript.

Thanks for the reviewer's suggestions. According to the reviewer's opinions, we have revised the whole paper and revised and replied to all points

Round 2

Reviewer 4 Report

The authors have revised the manuscript to the reviewer's satisfaction. It is recommended for publication.

Author Response

The English is not acceptable. New errors have occurred and lots of blanks are missing as a result of revision. Please, have the manuscript reviewed by a native speaker!

The grammar errors that appeared in the revised paper were corrected, and the blanks that appeared due to the revision were also corrected.

Page 5, Fig. 4a. The figure is dark except for the surface and the added lines. This is not acceptable for publication.

Figure 4a has been modified, and in this figure, composed of three different layers of materials, Q345 has a similar color to the zinc coating, but a distinct interface can still be seen

Page 6: The evaluation of porosity by use of Image J needs to be explained.

The Image J was used to make statistics on the porosity of the coating without and with CPS, and the results were shown in Figure 6. In image J software, the void is marked in red, and the porosity can be estimated by the ratio of the red area to the whole.

Figure 12 is not acceptable after revision. Either provide a better sketch or remove figure and associated text.

Considering that the paper mainly focuses on the microstructure and anti-corrosion properties of cold pressure sealing treatment, this part has been deleted.